# ORDERFUSION: ENCODING ORDERBOOK FOR END-TO-END PROBABILISTIC INTRADAY ELECTRICITY PRICE FORECASTING

## ABSTRACT

Probabilistic forecasting of intraday electricity prices is essential to manage market uncertainties. However, current methods rely heavily on domain feature extraction, which breaks the end-to-end training pipeline and limits the model's ability to learn expressive representations from the raw orderbook. Moreover, these methods often require training separate models for different quantiles, further violating the end-to-end principle and introducing the quantile crossing issue. Recent advances in time-series models have demonstrated promising performance in general forecasting tasks. However, these models lack inductive biases arising from buy–sell interactions and are thus overparameterized. To address these challenges, we propose an end-to-end probabilistic model called ORDERFUSION, which produces interaction-aware representations of buy–sell dynamics, hierarchically estimates multiple quantiles, and remains parameter-efficient with only **4,872** parameters. We conduct extensive experiments and ablation studies on price indices ($ID_1$, $ID_2$, and $ID_3$) using three years of orderbook in high-liquidity (German) and low-liquidity (Austrian) markets. The experimental results demonstrate that OrderFusion consistently outperforms multiple competitive baselines across markets, and ablation studies highlight the contribution of its individual components.

## 1 INTRODUCTION

In recent years, the rapid expansion of renewable energy has introduced significant variability and uncertainty in power generation due to weather dependence, resulting in power system imbalances Koch & Hirth (2019). The continuous intraday (CID) electricity market plays a pivotal role in mitigating imbalance challenges. Unlike traditional financial markets, where participants bid on future cash flows, traders in the CID electricity market submit bids and offers for electricity tied to specific delivery times Narajewski & Ziel (2020b). As a result, the CID electricity market significantly alleviates the demands on energy balance Ocker & Ehrhart (2017). In response to its growing importance and inherent uncertainty, increasing attention has been devoted to probabilistic electricity price forecasting. This task is particularly difficult due to complex trading behavior and the CID market's evolving dynamics, such as price jumps near delivery Lackes et al. (2025).

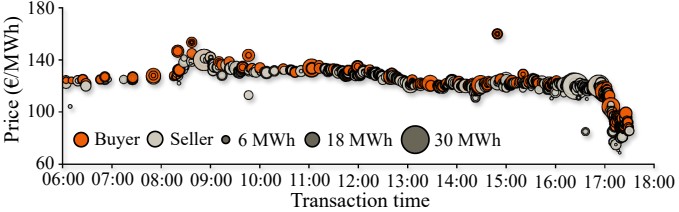

Figure 1: Buy-sell interactions for delivery at 18:00 on 2024-07-23. Buyers and sellers adjust bids and offers based on the opposite side, reflecting strategic interactions. As delivery time approaches, prices exhibit downward jumps.

Existing methods in probabilistic price forecasting rely heavily on domain feature extraction. Commonly used domain features, such as the Volume-Weighted Average Price (VWAP) over the past 15 minutes Marcjasz et al. (2020); Narajewski & Ziel (2020a); Serafin et al. (2022), directly aggregate over buy and sell sides, thereby overlooking buy-sell interactions Nagel (1995), shown in Figure

1. Other studies indicate that the last price already reflects past information, implying weak-form efficiency [1] Monteiro et al. (2016); Andrade et al. (2017); Janke & Steinke (2019); Uniejewski et al. (2019); Hirsch & Ziel (2024). However, relying solely on manual feature extraction not only breaks the end-to-end learning principle but also neglects the inductive biases arising from buy–sell interactions, requiring additional procedures for feature extraction and restricting the model from forming expressive representations from the raw orderbook.

Moreover, prior works on probabilistic price forecasting often require training separate models for each quantile. For instance, many adopt individual Linear Quantile Regression (LQR) models, where each quantile is predicted independently without shared representations Maciejowska & Nowotarski (2016); Serafin et al. (2019; 2022). This practice further violates the end-to-end learning principle and introduces quantile crossing, where higher quantile predictions fall below lower ones, leading to statistically invalid and unreliable predictive distributions Chernozhukov et al. (2010).

In recent years, advances in time-series modeling, such as FEDFormer Zhou et al. (2022), iTransformer Liu et al. (2023), PatchTST Nie et al. (2023), TimesNet Wu et al. (2023), and TimeXer Wang et al. (2024), have achieved notable success in general forecasting tasks by capturing complex temporal patterns. This suggests their potential applicability to intraday electricity price forecasting. However, these models lack mechanisms to incorporate the inductive bias arising from buy–sell interactions and typically require a large number of parameters to approximate such dynamics.

In this paper, we propose an end-to-end and parameter-efficient (4,872 parameters) model called OrderFusion, which produces interaction-aware representations of buy–sell dynamics and hierarchically estimates multiple quantiles via constrained residuals. We conduct extensive experiments and ablation studies on three widely used price indices ($ID_1$, $ID_2$, and $ID_3$) using three years of orderbook from the highly liquid German market, and further validate our approach on the less liquid Austrian market to assess its generalizability.

**Contributions**

- We propose and release OrderFusion, an end-to-end and parameter-efficient (4,872 parameters) probabilistic forecasting model tailored for CID electricity markets.

- We conduct experiments to compare OrderFusion against multiple baselines and examine its generalizability across markets with high (German) and low (Austrian) liquidity.

- We perform ablation studies to assess the impact of each architectural design choice, revealing the contribution of each component to overall predictive performance.

## 2 PRELIMINARY

The task is to forecast three widely used price indices: $ID_1$, $ID_2$, and $ID_3$, visualized in Figure 2. The forecast is made at time $t_f = t_d - \Delta$, with $t_d$ denoting the delivery time and $\Delta = 60 \times x$ min representing the lead time specific to price index $ID_x$, where $x \in \{1, 2, 3\}$. Each $ID_x$ is defined as the VWAP of trades executed within a specific time window before delivery:

$$ID_x = \frac{\sum\limits_{s \in S} \sum\limits_{t \in \mathcal{T}_f} P_t^{(s)} V_t^{(s)}}{\sum\limits_{s \in S} \sum\limits_{t \in \mathcal{T}_f} V_t^{(s)}}, \tag{1}$$

where the market side $s \in S = \{+, -\}$ corresponds to buy and sell orders, respectively, $t \in \mathcal{T}_f = [t_f, \ t_d - \delta_c]$ denotes the transaction time, $\mathcal{T}_f$ is the forecasting (trading) window, and $\delta_c$ is a market-specific parameter [2] Here, $P_t^{(s)}$ and $V_t^{(s)}$ denote the price and traded volume, respectively.

---

[1]Under the Efficient Market Hypothesis (EMH), a market is weak-form efficient if the recent prices reflect information contained in historical orders, such as past prices and volumes Fama (1970).

[2]For Germany, $\delta_c = 30$ min, and for Austria, $\delta_c = 0$ min. For other countries, $\delta_c$ can be retrieved from EPEX Spot download center under the category *Indices*.

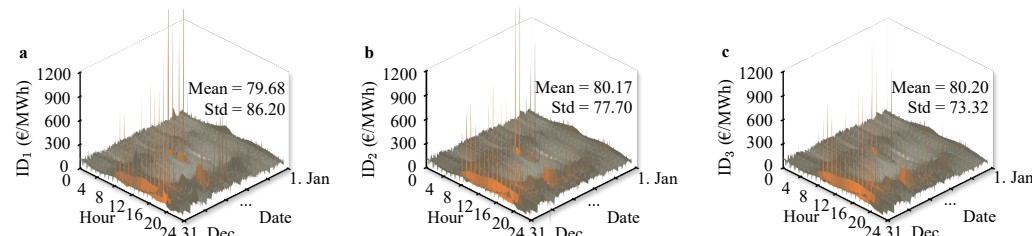

Figure 2: Distribution of prices over time and hour of day in the German market (2024). The second half of 2024 illustrates significant price spikes, indicating challenging forecasting tasks. Overall, volatility follows the order: $\text{ID}_1 > \text{ID}_2 > \text{ID}_3$. **(a)** $\text{ID}_1$ displays frequent price spikes, reflecting last-minute trading under imbalance pressure. **(b)** $\text{ID}_2$ reflects mid-session adjustments. **(c)** $\text{ID}_3$ corresponds to the most liquid trading window, exhibiting the least volatility.

## 3 MODEL

### 3.1 ENCODING

The encoding method separates the orderbook into two sides: buy $(+)$ and sell $(-)$. For each side, we treat all trades associated with each delivery time as one sample. Each trade contains a price and a traded volume. Additionally, we compute a relative time delta $\nabla t$ to encode temporal information:

$$\nabla t = t_d - t, \quad t < t_f. \tag{2}$$

Notably, the number of trades varies across samples, as trades are irregularly distributed over transaction time. Therefore, each sample is represented as a variable-length 2D sequence:

$$X_i^{(s)} = \begin{bmatrix} P_{t_1}^{(s)} & V_{t_1}^{(s)} & \nabla t_1 \\ P_{t_2}^{(s)} & V_{t_2}^{(s)} & \nabla t_2 \\ \vdots & \vdots & \vdots \\ P_{t_j}^{(s)} & V_{t_j}^{(s)} & \nabla t_j \\ \vdots & \vdots & \vdots \\ P_{t_{T_i^{(s)}}}^{(s)} & V_{t_{T_i^{(s)}}}^{(s)} & \nabla t_{T_i^{(s)}} \end{bmatrix} \tag{3}$$

where $X_i^{(s)} \in \mathbb{R}^{T_i^{(s)} \times 3}$ is the input matrix for the $i$-th sample on side $s$, with $T_i^{(s)}$ denoting the number of trades. The index $i \in \{1, 2, \ldots, N\}$ enumerates samples, and $j \in \{1, 2, \ldots, T_i^{(s)}\}$ denotes the $j$-th timestep within sample $i$. The encoded 2.5D representation consists of two irregular 2D sequences per sample for the buy and sell sides, respectively:

$$\mathcal{X}^{(+)} = \left\{ X_1^{(+)}, \ldots, X_i^{(+)}, \ldots, X_N^{(+)} \right\}, \quad \mathcal{X}^{(-)} = \left\{ X_1^{(-)}, \ldots, X_i^{(-)}, \ldots, X_N^{(-)} \right\} \tag{4}$$

### 3.2 BACKBONE

**Dual Masking Layer** As the number of matched trades varies between samples, we apply pre-padding to align all input sequences to a maximum length $T_{\max}$. Padding values are set to a constant $c = 10,000$ to ensure they do not occur in the data. Thus, the input dimension is standardized to $\mathbb{R}^{T_{\max} \times 3}$. To identify valid timesteps, we define a binary padding mask $\mathbf{B}_i^{(s)} \in \{0, 1\}^{T_{\max} \times 1}$ as:

$$\mathbf{B}_i^{(s)}[j] = \begin{cases} 1 & \text{if } X_i^{(s)}[j, :] \neq c, \\ 0 & \text{otherwise.} \end{cases} \tag{5}$$

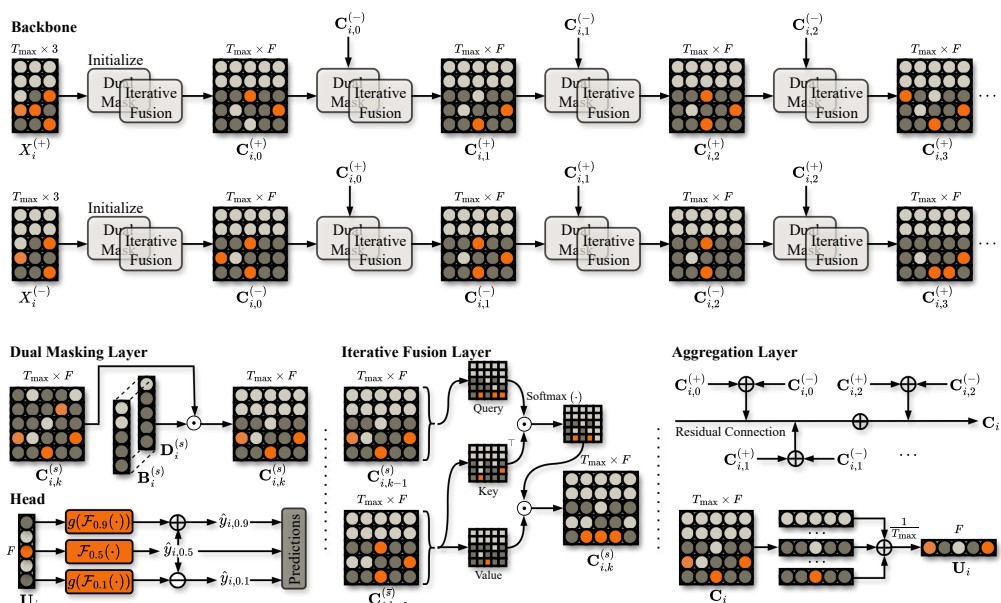

Figure 3: Structure of OrderFusion. The buy-side input and sell-side input are iteratively fused to form high-level representations of buy–sell interactions, which are then passed through a hierarchical head to generate multiple quantile estimates, enabling end-to-end probabilistic forecasting.

To reflect the prior that recent trades carry the most predictive information under the market efficiency hypothesis Hasbrouck (1991; 2007); Bacry et al. (2015), we define a binary temporal mask $\mathbf{D}_i^{(s)} \in \{0,1\}^{T_{\max} \times 1}$, where the cutoff length is given by $L = 2^\alpha$, controlled by a hyperparameter $\alpha \in \mathbb{N}$, with $L \leq T_{\max}$:

$$\mathbf{D}_i^{(s)}[j] = \begin{cases} 1 & \text{if } j > T_{\max} - L, \\ 0 & \text{otherwise.} \end{cases} \tag{6}$$

The dual mask is obtained by elementwise multiplication of the padding and temporal masks:

$$\mathbf{M}_i^{(s)} = \mathbf{B}_i^{(s)} \odot \mathbf{D}_i^{(s)}. \tag{7}$$

**Iterative Fusion Layer** As buyers and sellers iteratively adjust their bids and offers based on observed quotes from the opposite side, reflecting strategic interactions Nagel (1995), we design a series of iterative fusion layers to enable representation learning of such buy-sell interactions, illustrated in Figure 3:

$$\mathbf{C}_{i,k}^{(s)} = \begin{cases} X_i^{(s)} & \text{if } k = 0, \\ \mathbf{C}_{i,k-1}^{(s)} \mid \mathbf{C}_{i,k-1}^{(\bar{s})} & \text{if } k \geq 1, \end{cases} \tag{8}$$

where $k$ denotes the degree of interactions, and $\bar{s}$ is the opposite side of $s$. All intermediate representations are masked using $\mathbf{M}_i^{(s)}$ before being passed to subsequent layers.

For $k = 0$, the fusion representation is initialized by the masked input matrix $X_i^{(s)}$.

For $k \geq 1$, a cross-attention is applied, denoted by the fusion operator "$\mid$", where $\mathbf{C}_{i,k-1}^{(s)}$ serves as the query, and $\mathbf{C}_{i,k-1}^{(\bar{s})}$ serves as both the key and value:

$$\mathbf{Q}_{k-1}^{(s)} = \mathbf{C}_{i,k-1}^{(s)} \mathbf{W}_{\mathbf{Q},k-1}^{(s)}, \quad \mathbf{K}_{k-1}^{(\bar{s})} = \mathbf{C}_{i,k-1}^{(\bar{s})} \mathbf{W}_{\mathbf{K},k-1}^{(\bar{s})}, \quad \mathbf{V}_{k-1}^{(\bar{s})} = \mathbf{C}_{i,k-1}^{(\bar{s})} \mathbf{W}_{\mathbf{V},k-1}^{(\bar{s})}, \tag{9}$$

where $\mathbf{W}_{\mathbf{Q},k-1}^{(s)}, \mathbf{W}_{\mathbf{K},k-1}^{(\bar{s})}, \mathbf{W}_{\mathbf{V},k-1}^{(\bar{s})} \in \mathbb{R}^{F \times F}$ are learnable weights, and $F$ denotes the hidden dimension. The output of the cross-attention is computed as:

$$\mathbf{C}_{i,k}^{(s)} = \text{Softmax}\left(\frac{\mathbf{Q}_{k-1}^{(s)}(\mathbf{K}_{k-1}^{(\bar{s})})^\top}{\sqrt{F}}\right) \mathbf{V}_{k-1}^{(\bar{s})}. \tag{10}$$

Table 1: Examples and interpretations of representations. The table shows how side $s$ iteratively incorporates information from the opposite side $\bar{s}$, simulating buy-sell interactions.

| Degree | Representation | Interpretation |
|---|---|---|
| $k = 0$ | $\mathbf{C}_{i,0}^{(+)}$ | Buy-side |
| | $\mathbf{C}_{i,0}^{(-)}$ | Sell-side |
| $k = 1$ | $\mathbf{C}_{i,1}^{(+)} = \mathbf{C}_{i,0}^{(+)} \mid \mathbf{C}_{i,0}^{(-)}$ | Buy-side observed on sell-side |
| | $\mathbf{C}_{i,1}^{(-)} = \mathbf{C}_{i,0}^{(-)} \mid \mathbf{C}_{i,0}^{(+)}$ | Sell-side observed on buy-side |
| $k = 2$ | $\mathbf{C}_{i,2}^{(+)} = \mathbf{C}_{i,1}^{(+)} \mid \mathbf{C}_{i,1}^{(-)}$ | Evolved buy-side observed on evolved sell-side |
| | $\mathbf{C}_{i,2}^{(-)} = \mathbf{C}_{i,1}^{(-)} \mid \mathbf{C}_{i,1}^{(+)}$ | Evolved sell-side observed on evolved buy-side |
| $\ldots$ | $\ldots$ | $\ldots$ |

This allows side $s$ to observe the opposite side and form updated representations that reflect buy–sell interactions, illustrated in Table 1.

**Aggregation Layer** All of the fused representations at different degrees are combined via residual connection Xie et al. (2017) to produce the higher-level representation $\mathbf{C}_i \in \mathbb{R}^{T_{\max} \times F}$:

$$\mathbf{C}_i = \sum_{k=1}^{K} \left( \mathbf{C}_{i,k}^{(+)} + \mathbf{C}_{i,k}^{(-)} \right), \tag{11}$$

where $K$ denotes the maximum degree of interactions, and the summation is element-wise addition.

We apply the average pooling to obtain the attention-weighted average representation $\mathbf{U}_i \in \mathbb{R}^F$:

$$\mathbf{U}_i = \frac{1}{T_{\max}} \sum_{j=1}^{T_{\max}} \mathbf{C}_i[j]. \tag{12}$$

### 3.3 HEAD

The hierarchical head produces multiple quantile forecasts, where $\tau \in \mathcal{Q} = \{0.1, 0.5, 0.9\}$. In detail, the median quantile ($\tau = 0.5$) is learned from the shared representation $\mathbf{U}_i$ with one dense layer, denoted by $\mathcal{F}(\cdot)$:

$$\hat{y}_{i,0.5} = \mathcal{F}_{0.5}(\mathbf{U}_i). \tag{13}$$

For the upper quantile ($\tau = 0.9$), a residual is produced using $\mathbf{U}_i$ with another dense layer, which is enforced to be non-negative utilizing an absolute-value function $g(\cdot)$. The upper quantile prediction is then hierarchically computed by adding the non-negative residual to the median:

$$\hat{y}_{i,0.9} = \hat{y}_{i,0.5} + g(\mathcal{F}_{0.9}(\mathbf{U}_i)). \tag{14}$$

For the lower quantile ($\tau = 0.1$), a residual is similarly produced from $\mathbf{U}_i$ and enforced to be non-negative. The lower quantile prediction is then hierarchically computed by subtracting the non-negative residual from the median:

$$\hat{y}_{i,0.1} = \hat{y}_{i,0.5} - g(\mathcal{F}_{0.1}(\mathbf{U}_i)). \tag{15}$$

This design strictly enforces the ordering $\hat{y}_{i,0.1} \leq \hat{y}_{i,0.5} \leq \hat{y}_{i,0.9}$, overcoming the quantile crossing.

### 3.4 LOSS

Average Quantile Loss (AQL) is employed to jointly estimate multiple quantiles:

$$\text{AQL} = \frac{1}{N|\mathcal{Q}|} \sum_{i=1}^{N} \sum_{\tau \in \mathcal{Q}} L_\tau(y_i, \hat{y}_{i,\tau}), \tag{16}$$

where $y_i$ is the true price, $\hat{y}_i$ denotes the predicted price quantile, and the loss $L_\tau$ is defined as:

$$L_\tau(y_i, \hat{y}_{i,\tau}) = \begin{cases} \tau \cdot (y_i - \hat{y}_{i,\tau}), & \text{if } y_i \geq \hat{y}_{i,\tau}, \\ (1 - \tau) \cdot (\hat{y}_{i,\tau} - y_i), & \text{otherwise.} \end{cases} \tag{17}$$

When predicting upper quantiles, higher penalties are applied to under-predictions, whereas for lower quantiles, over-predictions incur higher penalties.

# 4 EXPERIMENT

We split the orderbook into training (2022-01-01 to 2024-01-01), validation (2024-01-01 to 2024-07-01), and testing (2024-07-01 to 2025-01-01). The choice of the testing period aims to include numerous extreme prices, as illustrated in Figure 2. We assess the model performance using the quantile losses ($Q_{0.1}$, $Q_{0.5}$, and $Q_{0.9}$), AQL, Average Quantile Crossing Rate (AQCR), Root Mean Squared Error (RMSE), Mean Absolute Error (MAE), and Coefficient of Determination ($R^2$). The Diebold-Mariano (DM) test is applied to determine if two models have a significant difference Diebold & Mariano (2002). All metrics are detailed in Appendix G. The hyperparameters are explained in Appendix H.

## 4.1 NAÏVE BASELINES

We include three seasonal naïve baselines: (i) **Naïve**[1] uses the price index from the most recent delivery hour; (ii) **Naïve**[2] uses the price index from the same delivery hour on the previous day; (iii) **Naïve**[3] uses the average price index from the same delivery hour over the past three days. To obtain probabilistic results, we compute empirical quantiles at individual levels ($\mathcal{Q} = \{0.1, 0.5, 0.9\}$).

The results from Table 2 show that OrderFusion significantly outperforms the naïve baselines, confirmed by both the probabilistic and pointwise DM tests, with all $p$-values $< 0.05$ and negative DM values. Compared to OrderFusion, the AQL averaged across price indices of Naïve[1] is 73.01% higher, while Naïve[2] and Naïve[3] are 361.74% and 359.67% higher, respectively, with an AQCR of 0.00%, as their forecasts are directly computed from historical values. The negative $R^2$ observed in Naïve[2] and Naïve[3] further suggests limited predictive performance.

## 4.2 DOMAIN-FEATURE-BASED METHODS

We include three domain-feature-based methods: (i) **15-Min VWAP:** prior studies report that the VWAP over the last 15 minutes is a strong domain feature; (ii) **Last Price:** existing studies indicate that the last price reflects past information, implying weak-form efficiency. We use this baseline to examine whether the CID market exhibits perfect weak-form efficiency; (iii) **Exhaustive Feature Set:** an extensive set of features is extracted, such as momentum, price percentiles, and traded volumes, totaling 384 features, detailed in Appendix F. To avoid model-specific bias, we evaluate both a deep learning model (MLP), and a statistical learning model (LQR).

The results in Table 2 show that OrderFusion outperforms all domain-feature-based methods, confirmed by both probabilistic and pointwise DM tests, with all $p$-values $< 0.05$ and negative DM values. Especially, the 21.65% improvement in AQL over the last price baseline suggests that the CID market is not perfectly weak-form efficient, and historical trades carry predictive information. Furthermore, the baseline with exhaustive feature sets leads to an average AQCR value of 0.15%, indicating unreliable probabilistic forecasts. This issue is expected to be magnified when predicting more quantiles. By design, OrderFusion consistently achieves an AQCR of 0.00%.

## 4.3 ADVANCED TIME-SERIES MODELS

We include five advanced time-series models as baselines: (i) **FEDFormer**, **iTransformer**, (iii) **PatchTST**, (iv) **TimesNet**, and (v) **TimeXer**. To apply these models, the masked buy-side and sell-side inputs are concatenated along the feature dimension.

The results in Table 2 show that OrderFusion outperforms all time-series baselines, as confirmed by the pointwise DM test. Notably, OrderFusion achieves 16.74% lower RMSE and 16.56% lower MAE, and improves $R^2$ by 0.04 relative to the mean of baselines. This performance gap is attributed to the fact that these models are designed for generic time-series tasks and lack the inductive biases of buy–sell interactions. Furthermore, while these baselines contain between 0.87M and 3.38M parameters, OrderFusion remains lightweight with only **4,872** parameters. We emphasize the importance of injecting the correct domain prior, instead of relying solely on stacking model parameters.

Table 2: Performance comparison on the German market. The superscript [1,2,3] denotes the inclusion of domain features (15-min VWAP, last price, and an exhaustive feature set), and only a better result between MLP and LQR is presented. The symbol "–" indicates that the model does not support probabilistic forecasting by design. All metrics are reported as mean±standard deviation over 5 independent runs. The best results are shown in **bold**, and the second-best results are underlined. The units of $Q_{0.1}$, $Q_{0.5}$, $Q_{0.9}$, AQL, RMSE, and MAE are expressed in €/MWh, and AQCR in %.

| Index | Model | $Q_{0.1}\downarrow$ | $Q_{0.5}\downarrow$ | $Q_{0.9}\downarrow$ | AQL $\downarrow$ | AQCR $\downarrow$ | RMSE $\downarrow$ | MAE $\downarrow$ | $R^2\uparrow$ |
|---|---|---|---|---|---|---|---|---|---|
| $ID_1$ | Naïve[1] | 5.70±0.00 | 8.99±0.00 | 5.64±0.00 | 6.78±0.00 | **0.00±0.00** | 45.97±0.00 | 18.00±0.00 | 0.74±0.00 |
| | Naïve[2] | 13.26±0.00 | 21.28±0.00 | 13.22±0.00 | 15.92±0.00 | **0.00±0.00** | 99.53±0.00 | 42.55±0.00 | -0.39±0.00 |
| | Naïve[3] | 12.80±0.00 | 21.04±0.00 | 13.67±0.00 | 15.84±0.00 | **0.00±0.00** | 94.62±0.00 | 42.09±0.00 | -0.10±0.00 |
| | MLP \| LQR[1] | 3.27±0.03 | 6.30±0.19 | 4.01±0.16 | 4.53±0.14 | 0.02±0.01 | 28.34±0.30 | 12.60±0.44 | 0.90±0.00 |
| | MLP \| LQR[2] | 3.23±0.08 | 6.71±0.13 | 3.99±0.14 | 4.64±0.10 | 0.01±0.00 | 53.55±0.95 | 13.42±0.43 | 0.65±0.01 |
| | MLP \| LQR[3] | 3.17±0.09 | 6.03±0.13 | 3.82±0.21 | 4.34±0.13 | 0.23±0.08 | 27.44±0.50 | 12.10±0.36 | 0.90±0.00 |
| | FEDFormer | – | – | – | – | – | 26.40±0.41 | 11.33±0.32 | 0.88±0.01 |
| | iTransformer | – | – | – | – | – | 27.02±0.47 | 11.25±0.28 | 0.88±0.01 |
| | PatchTST | – | – | – | – | – | 25.85±0.51 | 10.99±0.11 | 0.89±0.01 |
| | TimesNet | – | – | – | – | – | 25.24±0.28 | 10.73±0.24 | 0.89±0.01 |
| | TimeXer | – | – | – | – | – | 27.49±0.13 | 11.06±0.23 | 0.88±0.01 |
| | **OrderFusion** | **2.56±0.04** | **4.99±0.09** | **3.02±0.04** | **3.53±0.07** | **0.00±0.00** | **21.55±0.41** | **9.83±0.05** | **0.92±0.01** |
| $ID_2$ | Naïve[1] | 4.35±0.00 | 7.01±0.00 | 4.33±0.00 | 5.23±0.00 | **0.00±0.00** | 35.04±0.00 | 14.03±0.00 | 0.82±0.00 |
| | Naïve[2] | 11.99±0.00 | 19.64±0.00 | 12.32±0.00 | 14.65±0.00 | **0.00±0.00** | 89.58±0.00 | 39.27±0.00 | -0.19±0.00 |
| | Naïve[3] | 11.90±0.00 | 19.36±0.00 | 12.25±0.00 | 14.51±0.00 | **0.00±0.00** | 82.92±0.00 | 38.73±0.00 | -0.02±0.00 |
| | MLP \| LQR[1] | 3.05±0.07 | 5.56±0.11 | 3.84±0.09 | 4.15±0.10 | 0.01±0.00 | 34.29±0.43 | 11.12±0.21 | 0.83±0.00 |
| | MLP \| LQR[2] | 2.92±0.03 | 5.27±0.06 | 3.60±0.04 | 3.93±0.05 | 0.02±0.00 | 28.85±0.25 | 10.63±0.16 | 0.88±0.01 |
| | MLP \| LQR[3] | 2.53±0.02 | 5.26±0.07 | 3.59±0.03 | 3.80±0.05 | 0.21±0.05 | 26.00±0.23 | 9.40±0.17 | 0.89±0.01 |
| | FEDFormer | – | – | – | – | – | 27.55±0.44 | 10.03±0.25 | 0.84±0.01 |
| | iTransformer | – | – | – | – | – | 27.13±0.35 | 10.11±0.24 | 0.85±0.01 |
| | PatchTST | – | – | – | – | – | 26.80±0.42 | 9.97±0.25 | 0.86±0.02 |
| | TimesNet | – | – | – | – | – | 26.72±0.37 | 9.28±0.19 | 0.86±0.02 |
| | TimeXer | – | – | – | – | – | 27.04±0.69 | 10.17±0.48 | 0.85±0.01 |
| | **OrderFusion** | **2.13±0.06** | **4.20±0.05** | **2.66±0.04** | **2.99±0.05** | **0.00±0.00** | **21.64±0.23** | **8.26±0.18** | **0.91±0.01** |
| $ID_3$ | Naïve[1] | 3.91±0.00 | 6.39±0.00 | 3.87±0.00 | 4.72±0.00 | **0.00±0.00** | 28.49±0.00 | 12.78±0.00 | 0.87±0.00 |
| | Naïve[2] | 11.44±0.00 | 18.96±0.00 | 11.83±0.00 | 14.08±0.00 | **0.00±0.00** | 81.71±0.00 | 37.92±0.00 | -0.10±0.00 |
| | Naïve[3] | 11.61±0.00 | 18.87±0.00 | 11.82±0.00 | 14.10±0.00 | **0.00±0.00** | 77.57±0.00 | 37.74±0.00 | 0.01±0.00 |
| | MLP \| LQR[1] | 3.16±0.06 | 5.63±0.04 | 3.56±0.04 | 4.12±0.04 | 0.11±0.04 | 32.40±0.62 | 11.23±0.03 | 0.82±0.01 |
| | MLP \| LQR[2] | 2.65±0.05 | 5.31±0.02 | 3.37±0.03 | 3.77±0.03 | 0.01±0.00 | 33.44±0.65 | 10.62±0.28 | 0.83±0.02 |
| | MLP \| LQR[3] | 2.40±0.01 | 4.72±0.01 | 2.78±0.01 | 3.30±0.01 | 0.01±0.00 | 27.67±0.24 | 10.02±0.02 | 0.87±0.00 |
| | FEDFormer | – | – | – | – | – | 27.19±0.41 | 11.63±0.20 | 0.84±0.02 |
| | iTransformer | – | – | – | – | – | 27.78±0.58 | 11.79±0.40 | 0.85±0.02 |
| | PatchTST | – | – | – | – | – | 26.72±0.23 | 11.04±0.16 | 0.86±0.00 |
| | TimesNet | – | – | – | – | – | 26.70±0.14 | 10.83±0.12 | 0.85±0.01 |
| | TimeXer | – | – | – | – | – | 27.80±0.57 | 11.19±0.33 | 0.84±0.03 |
| | **OrderFusion** | **2.33±0.07** | **4.42±0.03** | **2.72±0.04** | **3.15±0.04** | **0.00±0.00** | **24.01±0.16** | **8.84±0.10** | **0.88±0.00** |

## 4.4 GENERALIZABILITY ASSESSMENT

To assess the generalizability of OrderFusion, we repeat all experiments on the less liquid Austrian market. Liquidity statistics are provided in Appendix D. The results are summarized in Table 3, and the same conclusions hold: OrderFusion outperforms all baselines. In detail, the AQL of Naïve[1], Naïve[2], and Naïve[3] is 56.65%, 208.54%, and 210.47% higher than that of OrderFusion, respectively. The extremely low or even negative $R^2$ of Naïve[2] and Naïve[3] further indicates poor forecasts. Relative to the last-price baseline, OrderFusion achieves 15.53% lower AQL, while maintaining an AQCR of 0.00%. Compared with the mean of the time-series baselines, OrderFusion yields 11.14% lower RMSE and 12.18% lower MAE, and improves $R^2$ by 0.03. These findings suggest that: (i) OrderFusion generalizes well to the less liquid market; (ii) the Austrian market is also not perfectly weak-form efficient; and (iii) modeling buy–sell interactions remains crucial.

## 5 ABLATION STUDY

### 5.1 DUAL MASKING LAYER

- **No Mask:** The Equation 7 is removed, and no masks are applied in iterative fusion layers.

Table 3: Performance comparison on the Austrian market.

| Index | Model | $Q_{0.1}\downarrow$ | $Q_{0.5}\downarrow$ | $Q_{0.9}\downarrow$ | AQL $\downarrow$ | AQCR $\downarrow$ | RMSE $\downarrow$ | MAE $\downarrow$ | $R^2\uparrow$ |
|---|---|---|---|---|---|---|---|---|---|
| $ID_1$ | Naïve[1] | 7.59±0.00 | 12.97±0.00 | 7.54±0.00 | 9.37±0.00 | **0.00±0.00** | 61.59±0.00 | 25.94±0.00 | 0.44±0.00 |
| | Naïve[2] | 12.57±0.00 | 21.15±0.00 | 12.40±0.00 | 15.37±0.00 | **0.00±0.00** | 94.80±0.00 | 42.31±0.00 | -0.32±0.00 |
| | Naïve[3] | 12.01±0.00 | 20.61±0.00 | 12.21±0.00 | 14.94±0.00 | **0.00±0.00** | 83.23±0.00 | 41.22±0.00 | -0.02±0.00 |
| | MLP \| LQR[1] | 5.43±0.08 | 10.18±0.10 | 5.33±0.11 | 6.98±0.10 | 0.00±0.00 | 42.02±0.54 | 20.37±0.21 | 0.74±0.00 |
| | MLP \| LQR[2] | 5.73±0.06 | 10.58±0.08 | 5.63±0.09 | 7.32±0.09 | 0.02±0.00 | 52.95±0.39 | 21.10±0.24 | 0.59±0.03 |
| | MLP \| LQR[3] | 5.24±0.23 | 10.08±0.38 | 5.38±0.11 | 6.91±0.37 | 0.25±0.05 | 40.06±0.63 | 19.53±0.10 | 0.76±0.00 |
| | FEDFormer | – | – | – | – | – | 39.92±0.93 | 19.93±0.65 | 0.74±0.02 |
| | iTransformer | – | – | – | – | – | 39.67±0.46 | 19.77±0.26 | 0.75±0.00 |
| | PatchTST | – | – | – | – | – | 38.15±1.03 | 19.86±0.74 | 0.75±0.01 |
| | TimesNet | – | – | – | – | – | 37.73±0.76 | 18.99±0.50 | 0.75±0.01 |
| | TimeXer | – | – | – | – | – | 38.12±1.27 | 19.30±0.88 | 0.74±0.01 |
| | OrderFusion | **4.46±0.08** | **8.55±0.09** | **4.34±0.04** | **5.77±0.08** | **0.00±0.00** | **32.93±0.32** | **16.79±0.14** | **0.78±0.00** |
| $ID_2$ | Naïve[1] | 4.93±0.00 | 8.30±0.00 | 4.89±0.00 | 6.04±0.00 | **0.00±0.00** | 38.40±0.00 | 16.59±0.00 | 0.67±0.00 |
| | Naïve[2] | 9.85±0.00 | 16.74±0.00 | 9.97±0.00 | 12.19±0.00 | **0.00±0.00** | 65.86±0.00 | 33.48±0.00 | 0.02±0.00 |
| | Naïve[3] | 10.04±0.00 | 17.15±0.00 | 10.28±0.00 | 12.49±0.00 | **0.00±0.00** | 62.09±0.00 | 34.30±0.00 | 0.13±0.00 |
| | MLP \| LQR[1] | 3.54±0.02 | 6.01±0.09 | 3.48±0.03 | 4.32±0.07 | 0.02±0.00 | 32.44±0.32 | 12.00±0.15 | 0.76±0.00 |
| | MLP \| LQR[2] | 3.35±0.07 | 5.98±0.04 | 3.35±0.02 | 4.24±0.06 | 0.01±0.00 | 30.84±0.20 | 11.98±0.13 | 0.78±0.00 |
| | MLP \| LQR[3] | 3.32±0.03 | 5.88±0.04 | 3.36±0.04 | 4.21±0.04 | 0.05±0.12 | 30.75±0.52 | 11.77±0.14 | 0.78±0.00 |
| | FEDFormer | – | – | – | – | – | 28.07±0.76 | 11.84±0.54 | 0.78±0.01 |
| | iTransformer | – | – | – | – | – | 29.51±0.49 | 12.04±0.17 | 0.77±0.02 |
| | PatchTST | – | – | – | – | – | 28.32±0.23 | 11.86±0.20 | 0.79±0.01 |
| | TimesNet | – | – | – | – | – | 28.00±0.22 | 11.78±0.18 | 0.79±0.01 |
| | TimeXer | – | – | – | – | – | 29.75±0.64 | 12.30±0.37 | 0.77±0.01 |
| | OrderFusion | **2.98±0.04** | **5.28±0.09** | **3.00±0.07** | **3.76±0.08** | **0.00±0.00** | **25.90±0.21** | **10.57±0.03** | **0.81±0.01** |
| $ID_3$ | Naïve[1] | 4.58±0.00 | 7.61±0.00 | 4.46±0.00 | 5.55±0.00 | **0.00±0.00** | 32.92±0.00 | 15.21±0.00 | 0.73±0.00 |
| | Naïve[2] | 9.47±0.00 | 16.08±0.00 | 9.59±0.00 | 11.71±0.00 | **0.00±0.00** | 59.68±0.00 | 32.16±0.00 | 0.12±0.00 |
| | Naïve[3] | 9.77±0.00 | 16.67±0.00 | 9.97±0.00 | 12.13±0.00 | **0.00±0.00** | 57.83±0.00 | 33.33±0.00 | 0.17±0.00 |
| | MLP \| LQR[1] | 3.42±0.02 | 5.97±0.03 | 3.42±0.02 | 4.27±0.02 | 0.01±0.00 | 28.81±0.07 | 11.94±0.05 | 0.80±0.00 |
| | MLP \| LQR[2] | 3.40±0.01 | 5.95±0.02 | 3.45±0.01 | 4.27±0.01 | 0.00±0.00 | 28.39±0.12 | 11.90±0.04 | 0.80±0.00 |
| | MLP \| LQR[3] | 3.23±0.01 | 5.92±0.01 | 3.42±0.01 | 4.20±0.01 | 0.01±0.00 | 28.33±0.23 | 11.69±0.06 | 0.80±0.00 |
| | FEDFormer | – | – | – | – | – | 28.76±0.31 | 11.73±0.40 | 0.77±0.01 |
| | iTransformer | – | – | – | – | – | 29.11±0.34 | 11.98±0.37 | 0.77±0.01 |
| | PatchTST | – | – | – | – | – | 28.44±0.62 | 11.76±0.27 | 0.79±0.00 |
| | TimesNet | – | – | – | – | – | 28.44±0.55 | 11.59±0.21 | 0.79±0.01 |
| | TimeXer | – | – | – | – | – | 29.02±1.06 | 11.89±0.44 | 0.76±0.02 |
| | OrderFusion | **3.11±0.05** | **5.42±0.05** | **3.06±0.07** | **3.84±0.05** | **0.00±0.00** | **25.72±0.12** | **10.69±0.11** | **0.81±0.00** |

- **Random Mask:** The Equation 7 is replaced with a randomly sampled vector, where each element is independently drawn from a uniform distribution over $[0, 1]$:

$$\mathbf{M}_i^{(s)} \sim \mathcal{U}(0, 1), \tag{18}$$

Results in Table 4 show that removing the mask leads to a 31.48% increase in AQL, as all padded values are treated as valid values, thereby introducing substantial noise. Randomizing the mask results in a 24.07% increase in AQL, emphasizing the importance of retaining only recent trades.

## 5.2 ITERATIVE FUSION LAYER

- **No Fusion:** The Equation 8 is removed. The buy- and sell-side inputs are directly passed to subsequent layers without representation learning of buy-sell interactions.

- **Self-Attention:** The buy- and sell-side inputs are concatenated along the feature dimension, and the concatenated input serves as query, key, and value. Thus, Equation 8 becomes:

$$\mathbf{C}_{i,k} = \begin{cases} X_i^{(+)} \| X_i^{(-)} & \text{if } k = 0, \\ \mathbf{C}_{i,k-1} \mid \mathbf{C}_{i,k-1} & \text{if } k \geq 1, \end{cases} \tag{19}$$

where $\|$ denotes concatenation along the feature dimension.

From Table 4, we observe that removing the iterative fusion layer results in an 18.52% increase in AQL. Replacing cross-attention with self-attention leads to a 6.79% higher AQL. These results further confirm that discarding the buy–sell inductive bias could degrade predictive performance.

Table 4: Ablation studies. The symbol $^\dagger$ marks the method used in OrderFusion.

| Method | $Q_{0.1} \downarrow$ | $Q_{0.5} \downarrow$ | $Q_{0.9} \downarrow$ | AQL $\downarrow$ | AQCR $\downarrow$ | RMSE $\downarrow$ | MAE $\downarrow$ | $R^2 \uparrow$ |
|---|---|---|---|---|---|---|---|---|
| No Mask | 2.86±0.31 | 6.28±1.32 | 3.65±0.60 | 4.26±0.72 | **0.00±0.00** | 34.92±6.85 | 12.56±1.63 | 0.75±0.08 |
| Random Mask | 2.92±0.24 | 5.44±0.78 | 3.67±0.23 | 4.02±0.45 | **0.00±0.00** | 32.45±4.20 | 10.85±0.84 | 0.79±0.04 |
| Dual Mask$^\dagger$ | **2.34±0.06** | **4.54±0.06** | **2.80±0.04** | **3.24±0.06** | **0.00±0.00** | **22.40±0.27** | **8.98±0.11** | **0.90±0.01** |
| No Fusion | 2.93±0.11 | 5.15±0.09 | 3.44±0.07 | 3.84±0.09 | **0.00±0.00** | 31.72±0.14 | 10.25±0.13 | 0.80±0.00 |
| Self-Attn. | 2.51±0.03 | 4.89±0.06 | 2.94±0.05 | 3.46±0.05 | **0.00±0.00** | 25.60±0.23 | 9.73±0.15 | 0.87±0.01 |
| Iter. Fusion$^\dagger$ | **2.34±0.06** | **4.54±0.06** | **2.80±0.04** | **3.24±0.06** | **0.00±0.00** | **22.40±0.27** | **8.98±0.11** | **0.90±0.01** |
| No Residual | 2.49±0.04 | 4.79±0.05 | 2.83±0.04 | 3.37±0.05 | **0.00±0.00** | 24.81±0.23 | 9.45±0.15 | 0.88±0.01 |
| Max Pool | 2.53±0.13 | 4.83±0.17 | 2.84±0.09 | 3.40±0.15 | **0.00±0.00** | 25.26±2.21 | 9.57±0.34 | 0.87±0.02 |
| Res. Conn. $^\dagger$ | **2.34±0.06** | **4.54±0.06** | **2.80±0.04** | **3.24±0.06** | **0.00±0.00** | **22.40±0.27** | **8.98±0.11** | **0.90±0.01** |
| Single-Q. Head | 2.41±0.08 | **4.52±0.06** | 2.82±0.04 | 3.25±0.06 | 1.17±0.71 | 22.67±0.40 | 9.15±0.13 | 0.90±0.01 |
| Post-Hoc Sort | 2.40±0.07 | 4.53±0.06 | 2.81±0.04 | 3.26±0.06 | **0.00±0.00** | 22.69±0.41 | 9.16±0.14 | 0.90±0.01 |
| Hier. Head$^\dagger$ | **2.34±0.06** | 4.54±0.07 | **2.80±0.04** | **3.24±0.06** | **0.00±0.00** | **22.40±0.27** | **8.98±0.11** | **0.90±0.01** |

## 5.3 AGGREGATION LAYER

- **No Residual Connection:** Only the representations with the maximum degree of interactions from Equation 11 are retained:

$$\mathbf{C}_i = \mathbf{C}_{i,K}^{(+)} + \mathbf{C}_{i,K}^{(-)}. \tag{20}$$

- **Max Pooling:** The average pooling in Equation 12 is replaced with max pooling:

$$\mathbf{U}_i = \max_{1 \leq j \leq T_{\max}} \mathbf{C}_i[j], \tag{21}$$

Results in Table 4 show that retaining only the representations with the maximum degree increases AQL value by 4.01%, as this operation loses the low-level features and leads to suboptimal performance. Replacing the average pooling with max pooling leads to a performance drop of 4.94% in AQL. Given that the prediction targets are VWAPs, average pooling offers a useful inductive bias.

## 5.4 HIERARCHICAL MULTI-QUANTILE HEAD

- **Single-Quantile Head:** The hierarchical multi-quantile head is replaced with a single-quantile head. Therefore, three models are trained independently for three quantiles.
- **Post-Hoc Sorting:** The predictions made by individual single-quantile models are re-ordered in ascending order Maciejowska & Nowotarski (2016); Serafin et al. (2019; 2022).

From Table 4, we observe that the single-quantile models achieve a comparable AQL value but suffer from quantile crossing with an AQCR of 1.17%. Although post-hoc sorting mitigates quantile crossing and yields an equivalent AQL, it introduces additional post-processing steps. In contrast, the hierarchical head eliminates quantile crossing, while maintaining the end-to-end design.

## 6 CONCLUSION

In this work, we propose OrderFusion, an end-to-end and parameter-efficient model, which consistently outperforms multiple baselines and generalizes across markets with both high (German) and low (Austrian) liquidity. The results reveal that CID electricity markets do not exhibit perfect weak-form efficiency, highlighting the value of historical trades. Our findings further underscore the importance of injecting the domain priors, rather than relying on stacking model parameters.

Despite strong performance, several limitations remain: (i) If CID markets evolve toward perfect weak-form efficiency, simple last-price models may suffice; (ii) OrderFusion is a deep learning model and lacks interpretability; and (iii) our experiments are constrained to Central Europe due to the high cost of orderbook (€23,400 for this study alone). Future work includes monitoring market status, exploring model interpretability, and expanding the dataset across European regions to support a pretrained foundation model.

**Ethics Statement**   We adhere to the ICLR Code of Ethics. Our study uses orderbook data from European electricity exchanges; no human subjects or personally identifiable information are involved. Data usage follows the providers' licenses. We release code and documentation for research purposes. We have no known conflicts of interest related to the data providers or outcomes reported.

**Reproducibility Statement**   The orderbook data used in this study is commercially available; its purchase source and data structure are described in Appendix B.1. To support reproducibility, we release well-documented code with an easy-to-use three-step pipeline, along with the code structure and usage guidelines in Appendix B.2. We report hardware and runtime details in Appendix C to facilitate realistic deployment of our proposed model. Additional reproducibility details are also provided, including data scaling (Appendix E), evaluation metrics (Appendix G), hyperparameter configurations and training procedures (Appendix H), and example forecasts (Appendix I).

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

## A    THE USE OF LARGE LANGUAGE MODELS (LLMS)

We employed GPT-4o to assist with grammar correction during the writing process. All LLM-generated suggestions were reviewed and edited to ensure they accurately reflect the authors' original intent. No content related to the methodology, analysis, or reference was generated by the LLM.

## B    CODE GUIDELINE

### B.1    DATA SOURCE

The orderbook data can be purchased from the EPEX Spot at: `https://webshop.eex-group.com/epex-spot-public-market-data`. Several data types are available. For example, the "Continuous Anonymous Orders History" for Germany costs € 325 per month.

The data structure is organized as follows:

```
|- Country (e.g. Germany)
    |- Intraday Continuous
        |- Orders
            |- Year (e.g. 2025)
                |- Month (e.g. 01)
                |- Month (e.g. 02)
                |- Month (e.g. 03)
                ...
        ...
```

### B.2    CODE USAGE

We open-source all code. The project directory is structured as follows:

```
OrderFusion/
    |- Data/
    |- Figure/
    |- Model/
    |- Result/
    |- OrderFusion.py
    |- Main.py
    |- Tutorial.ipynb
    |- README.md
```

The file `README.md` specifies the required package versions and dependencies. To facilitate reproducibility and accessibility, the full pipeline can be executed in three simple steps:

**Step 1:** Place the purchased orderbook data into the `Data/` folder.

**Step 2:** Execute `Main.py` to process the orderbook and obtain the optimized OrderFusion model. The script `OrderFusion.py` contains all utility functions.

**Step 3:** After execution, you can inspect: `Figure/` for visualizations (static images and GIFs) of forecasts versus true prices; `Model/` for saved model weights; `Result/` for evaluation metrics.

**Optional:** To better understand the code structure and functionality, run `Tutorial.ipynb`, which walks through the pipeline in three phases: `prepare` → `train` → `inference`.

## C    HARDWARE AND COMPUTATION

OrderFusion was evaluated on both an NVIDIA A100 GPU and an Intel Core i7-1265U CPU. The A100 targets high-performance computing, whereas the i7-1265U is a power-efficient processor commonly used in standard laptops. This comparison demonstrates the feasibility of deploying our model in environments without advanced GPU hardware. Training required approximately 1.5

minutes on the A100 and 6 minutes on the i7, while inference time was under 1 second in both cases, making the model suitable for continuous trading.

## D  MARKET LIQUIDITY AND PERFORMANCE

We assess the liquidity of the German and Austrian markets by examining the number of valid trades used as input. For example, when predicting $\text{ID}_1$, the average number of trades available as input is 2528.20 in the German market, whereas the Austrian market provides only 8.33% of that amount, indicating significantly lower liquidity. These averages are computed over the period from 2022-01-01 to 2025-01-01. Detailed statistics are summarized in Table 5. Notably, we observe that the liquidity correlates with the forecasting error, illustrated from Table 2 and Table 3. In detail, the AQL from Germany is consistently lower than that of Austria, indicating that more liquid markets yield lower forecasting errors. However, within the same country, the AQL does not follow the order $\text{ID}_1 < \text{ID}_2 < \text{ID}_3$, as the forecasting difficulty varies across indices. For example, $\text{ID}_3$ involves predictions made three hours in advance, whereas $\text{ID}_1$ corresponds to a shorter one-hour horizon. As a result, $\text{ID}_2$ represents a balanced trade-off, achieving the lowest AQL for both Germany and Austria.

Table 5: Number of input trades (mean±std).

| Market | Index | Number of Input Trades |
|--------|-------|------------------------|
| Germany | $\text{ID}_1$ | 2528.20±987.69 |
|         | $\text{ID}_2$ | 1579.21±699.60 |
|         | $\text{ID}_3$ | 1043.28±521.22 |
| Austria | $\text{ID}_1$ | 210.68±146.09 |
|         | $\text{ID}_2$ | 114.53±96.67 |
|         | $\text{ID}_3$ | 76.37±72.86 |

## E  DATA SCALING

To normalize the input features while ensuring robustness to outliers, we employ `RobustScaler` from `Scikit-Learn`. Unlike standard normalization methods that rely on the mean and standard deviation, `RobustScaler` centers each feature by subtracting its median and scales it using the interquartile range (IQR), defined as the difference between the 75th and 25th percentiles.

Since `RobustScaler` requires a 2D input matrix, we first vertically stack all input sequences. In detail, we stack the buy-side sequence $X_{i,\text{train}}^{(+)} \in \mathbb{R}^{T_i^{(+)} \times 3}$ and sell-side sequence $X_{i,\text{train}}^{(-)} \in \mathbb{R}^{T_i^{(-)} \times 3}$ across samples, resulting in a unified matrix $X_{\text{train}} \in \mathbb{R}^{\sum_{i=1}^{N_{\text{train}}}(T_i^{(+)} + T_i^{(-)}) \times 3}$. The scaler is fitted only on this aggregated matrix to prevent information leakage from the validation and test sets. After fitting, the transformation is applied to each sequence $X_{i,\text{set}}^{(s)}$ for $s \in \{+, -\}$ and set $\in \{\text{train}, \text{val}, \text{test}\}$. The full procedure is summarized in Algorithm 1.

Since price labels are naturally represented as 2D matrices, and thus `RobustScaler` can be directly applied without a customized procedure.

## F  EXHAUSTIVE FEATURE SET

### F.1  FEATURE EXTRACTION

We extract an exhaustive set of features from both the buy ($+$) and sell ($-$) sides across multiple look-back windows $\mathcal{T}_w = [t_f - \delta_w, t_f]$, where $\delta_w \in \{1, 5, 15, 60, 180, \infty\}$ (in minutes), and $\infty$ denotes the full available trading history. The full list of extracted features is summarized in Table 6. If no trades are recorded within a given window (e.g., $\delta_w = 1$), we fall back to the next longer window (e.g., $\delta_w = 5$) to extract features. If no trades are observed within the full history window ($\delta_w = \infty$),

---

**Algorithm 1** Customized Orderbook Feature Scaling

---

**Input:** Raw sequences $X_{i,\text{set}}^{(s)}$ for $s \in \{+, -\}$, set $\in \{\text{train}, \text{val}, \text{test}\}$
**Output:** Scaled sequences

Vertically stack all samples, timesteps, and sides in the training split:

$$X_{\text{train}} \leftarrow \texttt{Stack}\left(\bigcup_{i=1}^{N_{\text{train}}} \left[X_{i,\text{train}}^{(+)}; X_{i,\text{train}}^{(-)}\right]\right)$$

where $X_{i,\text{train}}^{(s)} \in \mathbb{R}^{T_i^{(s)} \times 3}$, and $X_{\text{train}} \in \mathbb{R}^{\sum_{i=1}^{N_{\text{train}}}(T_i^{(+)}+T_i^{(-)}) \times 3}$

Fit `RobustScaler` on $X_{\text{train}} \rightarrow$ `Scaler`

**for** each dataset split set $\in \{\text{train}, \text{val}, \text{test}\}$ **do**
    **for** each side $s \in \{+, -\}$ **do**
        **for** each sequence $X_{i,\text{set}}^{(s)}$ **do**
            $X_{i,\text{set}}^{(s)} \leftarrow$ `Scaler.transform`$(X_{i,\text{set}}^{(s)})$
        **end for**
    **end for**
**end for**
**return** Scaled $X_{i,\text{set}}^{(s)}$

---

the corresponding sample is discarded. Feature types include price and volume statistics (e.g., min, max, mean, percentiles), with percentile levels $p \in \mathcal{P} = \{10\%, 25\%, 45\%, 50\%, 55\%, 75\%, 90\%\}$.

## F.2 FEATURE SELECTION

The extracted feature set may contain redundant or noisy features that harm generalization. Following prior works in utilizing $\ell_1$-penalized linear regression, also known as Least Absolute Shrinkage and Selection Operator (LASSO), to encourage sparse feature sets for pointwise prediction Uniejewski et al. (2019), we extend this idea to the probabilistic forecasting setting by applying $\ell_1$-penalized Linear Quantile Regression (LQR). The hyperparameter $\alpha \in [1e-8, 1]$ controls the degree of sparsity by penalizing the absolute magnitudes of the coefficients. It is sampled at 100 evenly spaced values on a logarithmic scale and selected based on validation loss. Only features with non-zero coefficient magnitudes are retained, yielding a reduced sparse feature matrix.

## G METRICS

### G.1 QUANTILE LOSS AT INDIVIDUAL LEVELS

We compute quantile losses ($Q_{0.1}$, $Q_{0.5}$, and $Q_{0.9}$) separately for each target quantile:

$$Q_\tau = \frac{1}{N} \sum_{i=1}^{N} L_\tau\left(y_i, \hat{y}_{i,\tau}\right), \tag{22}$$

where $\tau \in \{0.1, 0.5, 0.9\}$.

### G.2 AVERAGE QUANTILE CROSSING RATE (AQCR)

AQCR captures the proportion of forecasted distributions that violate quantile monotonicity, i.e., when a lower quantile is predicted to be greater than a higher one. For each sample, the quantile crossing indicator is defined as:

$$C_i = \mathbb{I}\left(\max_{\tau_l < \tau_u}\left(\hat{y}_{i,\tau_l} - \hat{y}_{i,\tau_u}\right) > 0\right) \tag{23}$$

Table 6: Extracted features and definitions.

| Feature | Mathematical Definition |
|---|---|
| Price Percentile $\big|_{\mathcal{T}_w,\,p}^{(s)}$ | $\underset{t\in\mathcal{T}_w,\,p}{\text{percentile }} P_t^{(s)}$ |
| Min Price $\big|_{\mathcal{T}_w}^{(s)}$ | $\underset{t\in\mathcal{T}_w}{\min}\, P_t^{(s)}$ |
| Max Price $\big|_{\mathcal{T}_w}^{(s)}$ | $\underset{t\in\mathcal{T}_w}{\max}\, P_t^{(s)}$ |
| First Price $\big|_{\mathcal{T}_w}^{(s)}$ | $\underset{t\in\mathcal{T}_w}{\text{first}}\, P_t^{(s)}$ |
| Last Price $\big|_{\mathcal{T}_w}^{(s)}$ | $\underset{t\in\mathcal{T}_w}{\text{last}}\, P_t^{(s)}$ |
| Mean Price $\big|_{\mathcal{T}_w}^{(s)}$ | $\bar{P}_{\mathcal{T}_w}^{(s)}$ |
| Price Volatility $\big|_{\mathcal{T}_w}^{(s)}$ | $\sqrt{\dfrac{1}{n_{\mathcal{T}_w}^{(s)}}\sum_{t\in\mathcal{T}_w}\big(P_t^{(s)}-\bar{P}_{\mathcal{T}_w}^{(s)}\big)^2}$ |
| Delta Price $\big|_{\mathcal{T}_w}^{(s)}$ | $\underset{t\in\mathcal{T}_w}{\text{last}}\, P_t^{(s)} - \underset{t\in\mathcal{T}_w}{\text{first}}\, P_t^{(s)}$ |
| Volume Percentile $\big|_{\mathcal{T}_w,\,p}^{(s)}$ | $\underset{t\in\mathcal{T}_w,\,p}{\text{percentile }} V_t^{(s)}$ |
| Min Volume $\big|_{\mathcal{T}_w}^{(s)}$ | $\underset{t\in\mathcal{T}_w}{\min}\, V_t^{(s)}$ |
| Max Volume $\big|_{\mathcal{T}_w}^{(s)}$ | $\underset{t\in\mathcal{T}_w}{\max}\, V_t^{(s)}$ |
| First Volume $\big|_{\mathcal{T}_w}^{(s)}$ | $\underset{t\in\mathcal{T}_w}{\text{first}}\, V_t^{(s)}$ |
| Last Volume $\big|_{\mathcal{T}_w}^{(s)}$ | $\underset{t\in\mathcal{T}_w}{\text{last}}\, V_t^{(s)}$ |
| Mean Volume $\big|_{\mathcal{T}_w}^{(s)}$ | $\bar{V}_{\mathcal{T}_w}^{(s)}$ |
| Volume Volatility $\big|_{\mathcal{T}_w}^{(s)}$ | $\sqrt{\dfrac{1}{n_{\mathcal{T}_w}^{(s)}}\sum_{t\in\mathcal{T}_w}\big(V_t^{(s)}-\bar{V}_{\mathcal{T}_w}^{(s)}\big)^2}$ |
| Delta Volume $\big|_{\mathcal{T}_w}^{(s)}$ | $\underset{t\in\mathcal{T}_w}{\text{last}}\, V_t^{(s)} - \underset{t\in\mathcal{T}_w}{\text{first}}\, V_t^{(s)}$ |
| Sum Volume $\big|_{\mathcal{T}_w}^{(s)}$ | $\sum_{t\in\mathcal{T}_w} V_t^{(s)}$ |
| Trade Count $\big|_{\mathcal{T}_w}^{(s)}$ | $n_{\mathcal{T}_w}^{(s)}$ |
| VWAP $\big|_{\mathcal{T}_w}^{(s)}$ | $\dfrac{\sum_{t\in\mathcal{T}_w} P_t^{(s)} V_t^{(s)}}{\sum_{t\in\mathcal{T}_w} V_t^{(s)}}$ |
| Momentum $\big|_{\mathcal{T}_w}^{(s)}$ | $\dfrac{\underset{t\in\mathcal{T}_w}{\text{last}}\, P_t^{(s)} - \text{VWAP}^{(s)}}{\text{VWAP}^{(s)}}$ |

where $\mathbb{I}(\cdot)$ is an indicator function that returns 1 if any quantile pair fulfills the condition inside and 0 otherwise.

We compute the AQCR as:

$$\text{AQCR} = \frac{1}{N}\sum_{i=1}^{N} C_i. \tag{24}$$

A lower AQCR indicates fewer quantile crossing violations and thus reflects more reliable probabilistic forecasts.

### G.3 ROOT MEAN SQUARED ERROR (RMSE)

The RMSE evaluates the accuracy of pointwise predictions by penalizing larger errors more heavily than smaller ones. It is particularly sensitive to outliers and provides an overall measure of prediction quality. RMSE is calculated as:

$$\text{RMSE} = \sqrt{\frac{1}{N} \sum_{i=1}^{N} (y_i - \hat{y}_i)^2}, \tag{25}$$

where $y_i$ represents the true value, $\hat{y}_i$ is the predicted value, and $N$ is the total number of samples.

### G.4 MEAN ABSOLUTE ERROR (MAE)

The MAE measures the average magnitude of prediction errors, treating all deviations equally regardless of their direction. Unlike RMSE, MAE is more robust to outliers, making it a reliable metric for assessing average prediction accuracy. It is computed as:

$$\text{MAE} = \frac{1}{N} \sum_{i=1}^{N} |y_i - \hat{y}_i|, \tag{26}$$

where $y_i$ and $\hat{y}_i$ are the true and predicted values, respectively.

### G.5 COEFFICIENT OF DETERMINATION

The Coefficient of Determination ($R^2$) quantifies the proportion of variance in the target variable that is explained by the predictions. A value of $R^2 = 1$ indicates perfect predictions, whereas $R^2 = 0$ suggests that the model performs no better than predicting a mean value. It is defined as:

$$R^2 = 1 - \frac{\sum_{i=1}^{N} (y_i - \hat{y}_i)^2}{\sum_{i=1}^{N} (y_i - \bar{y})^2}, \tag{27}$$

where $\bar{y}$ is the mean of the true values $y_i$, and the numerator and denominator represent the residual sum of squares and the total sum of squares, respectively.

### G.6 DIEBOLD & MARIANO (DM) TEST

To assess if differences in forecasting performance are statistically significant, the DM test is applied.

For probabilistic forecasts, we compute the loss differential at each quantile $\tau \in \mathcal{Q}$ between two models $l \in \{1, 2\}$:

$$\text{diff}_{i,\tau} = L_\tau \left( y_i, \hat{y}_{i,\tau}^{(1)} \right) - L_\tau \left( y_i, \hat{y}_{i,\tau}^{(2)} \right). \tag{28}$$

For point forecasts, the loss differential between two models is computed for each sample as:

$$\text{diff}_i = \left| y_i - \hat{y}_{i,0.5}^{(1)} \right| - \left| y_i - \hat{y}_{i,0.5}^{(2)} \right|. \tag{29}$$

The DM test statistic is then calculated as:

$$\text{DM} = \frac{\bar{\text{diff}}}{\hat{\sigma}_{\text{diff}} / \sqrt{M}}, \tag{30}$$

$$\bar{\text{diff}} = \frac{1}{M} \sum_{j=1}^{M} \text{diff}_j, \tag{31}$$

where $M = N \cdot |\mathcal{Q}|$ for probabilistic forecasts, and $M = N$ for point forecasts. The index $j$ enumerates all prediction instances across dimensions, and $\hat{\sigma}_{\text{diff}}$ is the sample standard deviation of $\{\text{diff}_j\}_{j=1}^{M}$. We compute a $p$-value; if $p < 0.05$ and the DM value is positive (negative), then we report that model 2 (model 1) significantly outperforms the other in Section 4. The rules are summarized in Table 7.

Table 7: Interpretation of DM test outcomes.

| Condition | Interpretation | Conclusion |
|---|---|---|
| $p < 0.05$, DM $> 0$ | Statistically significant | Model 2 is better |
| $p < 0.05$, DM $< 0$ | Statistically significant | Model 1 is better |
| $p \geq 0.05$ | Not statistically significant | – |

## H    HYPERPARAMETER OPTIMIZATION

The models were optimized based on validation loss through an exhaustive grid search, and the best model with the lowest validation loss was saved. The search space of key hyperparameters is listed in Table 8. Each model was repeatedly trained 5 times using the optimal hyperparameters to report the mean and standard deviation in Table 2 and Table 3.

**Domain-Feature-Based Methods:**   For models with 15-min VWAP and last price features, the $\ell_1$ regularization from LQR was not tuned as it involved only a single feature. For the exhaustive feature set, $\ell_1$ was optimized to obtain a sparse feature set.

**Advanced Time-Series Models:**   For FEDFormer, iTransformer, PatchTST, TimesNet, and TimeXer, the invalid combinations of hyperparameters were skipped; for example, when `hidden_size = 4` and `n_heads = 8`, the resulting dimension of a single attention head would be a non-integer, which is not permissible. If not specified in Table 8, recommended hyperparameters are used from the original paper.

**OrderFusion:**   The optimizer used in OrderFusion is Adam. The number of training epochs was set to 50, the batch size to 256, and the learning rate to $4 \times 10^{-3}$ with exponential decay by a factor of 0.95 every 10 epochs. After obtaining the optimal hyperparameters from Table 8, we empirically varied the learning rate to 1e-3 and 7e-3, and the batch size to 64 and 1024, and observed that a similarly low validation and testing loss could always be reached within 50 epochs, suggesting that the model is not sensitive to slight changes in learning rate and batch size with the optimized hyperparameters. Notably, the optimal `cutoff_length` varied across price indices, showing that the effective order depth differs by price indices and suggesting that the market is not perfectly weak-form efficient. Moreover, we do not observe any significant AQL difference when setting $T_{\max}$ to a fixed value of 512 or 128, compared to using the maximum number of trades across all samples. However, when reducing $T_{\max}$ to 8 or smaller, the AQL increases accordingly, potentially because these distant trades still carry predictive information. We recommend setting $T_{\max} = 128$, as this value does not significantly increase computational cost while providing a buffer against market changes. Furthermore, the optimal `interaction_degree = 4` indicates that buy–sell interactions exhibit an intermediate level of complexity, where more complicated features are unnecessary.

## I    EXAMPLE FORECASTING

Figure 4 and Figure 5 visualize the predicted quantiles against the ground truth under normal and extreme price conditions, respectively. Overall, OrderFusion delivers accurate probabilistic forecasts in normal price regimes and satisfactory performance under extreme conditions. More forecasting examples are visualized as GIF files in the supplementary material.

## J    NAIVE TRADING STRATEGIES

### J.1    POINTWISE FORECASTS

With pointwise forecasts, a single predicted value (typically the median or mean) is used to inform trading decisions. A simple strategy is to buy if the predicted VWAP is higher than the current market price, anticipating a profitable upward move. Alternatively, the predicted VWAP can be compared to a recent average price to decide whether market conditions are improving. These strategies are easy to implement but ignore uncertainty, leading to suboptimal decisions under volatile conditions.

Table 8: Hyperparameter search space.

| Model | Search Space |
|---|---|
| LQR | $\ell_1$: {5e-8, 1e-8, 5e-7, 1e-7, ..., 1} |
| MLP | hidden_size: {4, 16, 64, 256, 512}
n_layers: {1, 2, 4, 8}
dropout: {0.1, 0.2, 0.4} |
| FEDFormer | hidden_size: {4, 16, 64, 256, 512}
conv_hidden_size: {8, 32, 128}
n_layers: {1, 2, 4, 8}
n_heads: {1, 2, 4, 8}
moving_window: {4, 16, 64} |
| iTransformer | hidden_size: {4, 16, 64, 256, 512}
n_layers: {1, 2, 4, 8}
n_heads: {1, 2, 4, 8}
d_ff: {512, 1024, 2048}
dropout: {0.1, 0.2, 0.4} |
| PatchTST | hidden_size: {4, 16, 64, 256, 512}
n_layers: {1, 2, 4, 8}
n_heads: {1, 2, 4, 8}
patch_len: {4, 8, 16}
dropout: {0.1, 0.2, 0.4, 0.8} |
| TimesNet | hidden_size: {4, 16, 64, 256, 512}
conv_hidden_size: {8, 32, 128}
n_layers: {1, 2, 4, 8}
top_k: {1, 2, 4, 8} |
| TimeXer | hidden_size: {4, 16, 64, 256, 512}
n_layers: {1, 2, 4, 8}
n_heads: {1, 2, 4, 8}
d_ff: {64, 256, 1024} |
| OrderFusion | hidden_size: {4, 16, 64, 256, 512}
cutoff_length: {1, 4, 16, 64, 256}
interaction_degree: {1, 2, 4, 8} |

### J.2 PROBABILISTIC FORECASTS

Probabilistic forecasts provide a range of quantiles, enabling strategies that account for prediction uncertainty. One conservative strategy is to buy only if the 10th percentile of the predicted VWAP exceeds the current market price, ensuring that even in a pessimistic scenario, the trade is expected to be profitable. A more optimistic strategy buys if the 90th percentile is above a recent average price, targeting trades with high upside potential. These approaches go beyond single-point estimates by incorporating confidence levels, allowing more informed and risk-sensitive trading decisions.

## K IMPACT STATEMENT

OrderFusion is directly applicable to all European CID electricity markets, as the orderbook data format provided by EPEX Spot is standardized across regions. By generating accurate probabilistic forecasts, our model supports the development of risk-aware bidding strategies that are essential for market participants, such as traders and aggregators, to navigate volatile intraday price dynamics. Ultimately, OrderFusion facilitates a smoother transition toward renewable energy integration.

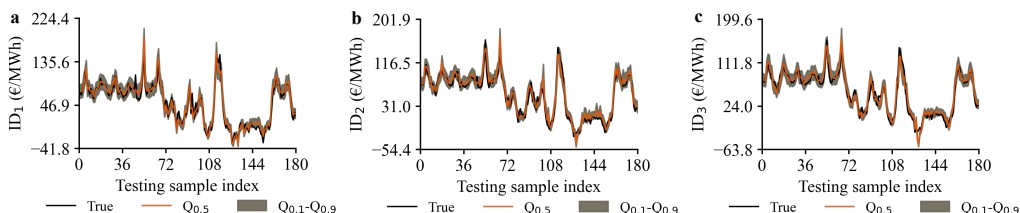

Figure 4: Visualization of example forecasts on the testing set in normal price regime. **(a)** $ID_1$ forecast. **(b)** $ID_2$ forecast. **(c)** $ID_3$ forecast. The plots show the true prices (black), median prediction $Q_{0.5}$ (orange), and the 80% prediction interval between $Q_{0.1}$ and $Q_{0.9}$ (gray band). The close alignment between predicted and true prices across different horizons demonstrates accurate probabilistic forecasts of OrderFusion in normal price regimes.

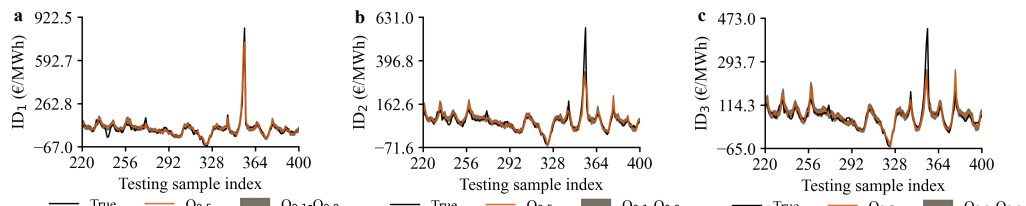

Figure 5: Visualization of example forecasts on the testing set in extreme price regime. **(a)** $ID_1$ forecast. **(b)** $ID_2$ forecast. **(c)** $ID_3$ forecast. Despite sharp price spikes and high volatility, the model captures the overall trend and maintains coherent prediction intervals. All three subplots show extreme prices occurring at the same sample index, indicating that price spikes are concentrated in the last hour before delivery. Predicting these extremes several hours in advance—particularly for $ID_3$—is more challenging than for shorter horizons. Nonetheless, the results demonstrate that OrderFusion delivers satisfactory performance even under extreme market conditions.

