# OpenReview forum: "OrderFusion: Encoding Orderbook for End-to-End Probabilistic Intraday Electricity Price Forecasting"
_ICLR.cc/2026/Conference — ICLR 2026 Conference Withdrawn Submission_

### Official Review · Reviewer_Zofn · 2025-10-29

**Soundness:** 2
**Presentation:** 3
**Contribution:** 2
**Rating:** 2
**Confidence:** 3

**Summary:**

This paper proposes ORDERFUSION, a method designed to perform intraday electricity price forecasting by encoding orderbooks. Experiments show that the proposed method performs competitively against comparing methods with a small number (~5,000) of parameters.

**Strengths:**

- The proposed method demonstrates competitive performance
- Ablation study is performed, showing the effectiveness of individual components

**Weaknesses:**

- It has been claimed that "prior works on probabilistic price forecasting often require training separate models for each quantile" and "they introduce quantile crossing". However, in the general forecasting setting, there are long-established and well-known probabilistic methods (see, e.g., [1-2]) that models multiple quantile levels using only one model. These works have already proposed to use techniques very similar to "Section 3.3 Head" of this paper to avoid quantile crossing. In addition, unlike the proposed approach, the existing approaches can model all quantile levels. Introducing existing techniques from general forecasting to the specific setting of price forecasting has a limited novelty.

- According to Table 5, it seems that the experimental data are of small scale. And the proposed model is also of small scale (5,000 parameters). However, the state-of-the-art comparing methods are big Transformer models. It is unclear whether the performance gain is from the effective design of the model components, or from the potential overfitting of the comparing methods and/or the inductive bias coming from the small size of the proposed model.

- Although the model has achieved competitive performance on intraday electricity price forecasting by utilizing the nature of orderbooks, I can't help but think that the application is too specific. I think this paper might be better fitted to submit to a price forecasting journal rather than ICLR.

[1] Gasthaus, J., Benidis, K., Wang, Y., Rangapuram, S. S., Salinas, D., Flunkert, V., & Januschowski, T. (2019, April). Probabilistic forecasting with spline quantile function RNNs. In The 22nd international conference on artificial intelligence and statistics (pp. 1901-1910). PMLR.

[2] Park, Y., Maddix, D., Aubet, F. X., Kan, K., Gasthaus, J., & Wang, Y. (2022, May). Learning quantile functions without quantile crossing for distribution-free time series forecasting. In International conference on artificial intelligence and statistics (pp. 8127-8150). PMLR.

**Questions:**

See weaknesses

---

### Official Review · Reviewer_uPXG · 2025-10-29

**Soundness:** 2
**Presentation:** 3
**Contribution:** 2
**Rating:** 2
**Confidence:** 5

**Summary:**

The paper addresses probabilistic forecasting of intraday electricity prices. It introduces OrderFusion, an end-to-end model designed to (1) avoid quantile crossing and (2) leverage buy–sell order dynamics for improved accuracy.

Experiments are conducted on German and Austrian markets, but the baselines (MLP and naïve methods) are not specialized for this task.

**Strengths:**

1. The paper includes comparisons with well-established naïve baselines, but the set of specialized EPF baselines is limited. The authors should refer to “Forecasting Day-Ahead Electricity Prices: A Review of State-of-the-Art Algorithms, Best Practices and an Open-Access Benchmark.”

2. Comprehensive ablation studies are presented, yet their relevance is undermined by the weak choice of baseline models.

**Weaknesses:**

1. A 4K-parameter model is too small to learn hierarchical representations from data, relying mainly on raw input features. ICLR typically values representation learning capabilities.

2. The model is only tested on electricity price data, which limits generality. ICLR favors broadly applicable methods. Venues like Energy Economics or Applied Energy may be a better fit for this paper.

3. The “advanced” baselines (FEDFormer, iTransformer, TimesNet, TimeXer) do not output quantiles and are thus unsuitable for probabilistic forecasting. More appropriate baselines include probabilistic versions available in NeuralForecast or GluonTS (e.g., NBEATSx (EPF), NHITS).

4. The test set spans only six months (2024-07–2025-01), missing full-year seasonality and holiday effects critical in EPF. A longer test period is recommended. See “Forecasting Day-Ahead Electricity Prices: A Review of State-of-the-Art Algorithms, Best Practices and an Open-Access Benchmark.”.

**Questions:**

1. Equation (1) has a subindex "x", however nothing within the equation reflects the subindex "x". Why?

2. Why is the selection of probabilistic forecasting baselines so poor? FedFormer, iTransformer, TimesNet are clearly not specialized in probabilistic EPF.

3. Would the hierarchical head that prevents quantile crossing be a nice addition to other forecasting architectures?

---

### Official Review · Reviewer_hw2E · 2025-10-31

**Soundness:** 3
**Presentation:** 3
**Contribution:** 3
**Rating:** 6
**Confidence:** 4

**Summary:**

The paper introduces OrderFusion, a tiny (about 4872 parameters) end-to-end model for probabilistic intraday electricity price forecasting on raw order-book trades. It encodes buy/sell streams separately, applies a dual mask (padding + recent-trade prior), then learns buy–sell interaction representations via iterative cross-attention (“fusion”) before a hierarchical multi-quantile prediction head predicts. Experiments on three indices (ID1/ID2/ID3) across Germany (high liquidity) and Austria (lower liquidity) show consistent gains over naïve, domain-feature, and modern TS models; ablations attribute improvements to the mask, the iterative fusion, residual aggregation, and the hierarchical head.

**Strengths:**

1. Originality. Iterative cross-side fusion directly encodes strategic buy–sell reactions; the dual mask injects a recency prior without heavy architectures; the hierarchical head enforces quantile ordering end-to-end.
2. Quality. Proper temporal split, diverse metrics, DM tests, two distinct markets, and thorough ablations (mask, fusion, pooling, head) provide a credible empirical foundation.
3. Clarity. The architecture figure and equations make it easy to follow. The task/indices are well defined.
4. Significance. Strong, consistent wins vs. domain-feature baselines and heavy TS models, with huge parameter savings. Evaluations on CPUs suggests deployability.

**Weaknesses:**

1. Results are limited to Germany and Austria, which are both in Europe.
2. The paper did not compare its model to LOB deep learning models. It only evaluated the general-purpose TS models.

**Questions:**

Did you tune mask length L per index or per market, and how robust is performance across L?

---

### Official Review · Reviewer_Xyg9 · 2025-10-31

**Soundness:** 2
**Presentation:** 3
**Contribution:** 2
**Rating:** 4
**Confidence:** 4

**Summary:**

This paper introduces OrderFusion, an end-to-end deep learning model for probabilistic forecasting of intraday electricity prices. It addresses some issues of the existing methods, such as the need for manual feature extraction and the quantile crossing problem, where predicted quantiles can become inconsistent. OrderFusion uses an Iterative Fusion Layer with cross-attention to model buy-sell interactions on the exchange platform and a hierarchical head to predict multiple quantiles in the correct order. Despite having only around 5k parameters, the model outperforms forecasting methods on German and Austrian market data.

**Strengths:**

Unlike general time-series models that use too many parameters, OrderFusion directly models buy-sell interactions. It does this using an Iterative Fusion Layer with cross-attention, which mimics how buyers and sellers adjust their orders in response to each other.

Instead of training separate models for each quantile, which can cause inconsistent results, OrderFusion uses a hierarchical head that first predicts the median and then estimates upper and lower quantiles as non-negative differences from it.

Also, the paper is well-written, clearly structured, and easy to follow.

**Weaknesses:**

The empirical evaluation is limited, as OrderFusion is compared only with naïve baselines. It should also be bench-marked against recent probabilistic models like DeepAR, lag-llama or  TFT for a fairer comparison.

Since upper and lower quantiles are derived from the median, any error in the median prediction propagates to all quantiles, reducing overall accuracy.

Enforcing non-negative residuals ensures that predicted quantiles (e.g., 0.1, 0.5, 0.9) never cross, but it also fixes the spacing between them, limiting flexibility in modeling varying uncertainty. For instance, during stable market periods, the gap between quantiles should be narrow, while during volatile periods it should widen. However, this constraint forces a similar spacing in both cases, leading the model to underestimate uncertainty when volatility spikes and overestimate it when prices are stable.

The model’s performance cannot be verified because the dataset used is not publicly available. This is an important in our view.

The authors should have tried with a openly available real data-set. Else, they could have generated a comparable synthetic/simulated dataset to validate the model’s performance and ensure reproducibility.

**Questions:**

The model should be compared with more recent probabilistic forecasting approaches to ensure a fair and comprehensive performance evaluation.

Since the dataset used is commercially restricted, the model’s performance cannot be independently verified. The authors should simulate a comparable dataset to support reproducibility.

Need to address 3rd point of weakness

---

### Note · Authors · 2025-12-03

I have read and agree with the venue's withdrawal policy on behalf of myself and my co-authors.